

# Evaluating Airborne Ku-Band Radar Altimetry over Landfast First-Year Sea Ice

Paul Donchenko[1], Joshua King[2], and Richard Kelly[1]

[1]Department of Geography, University of Waterloo, Waterloo, ON N2L 3G1, Canada
[2]Climate Research Division, Environment and Climate Change Canada, Toronto, M3H5T4, Canada
**Correspondence:** Joshua King (joshua.king@canada.ca)

**Abstract.**

Recent studies have challenged the assumption that Ku-band radar used by the CryoSat-2 altimeter fully penetrates the dry snow cover of Arctic sea ice. There is also uncertainty around the proper technique for handling retracker threshold selection in the Threshold First-Maxima Retracker (TFMRA) method which estimates the ice surface elevation from the radar echo
waveform. The purpose of this study was to evaluate the accuracy and penetration of the TFMRA retracking method applied to the Airborne Synthetic Aperture Radar and Interferometric Radar Altimeter System (ASIRAS), an airborne simulator of the CryoSat-2, to investigate the effect of surface characteristics and improve accuracy.

The ice surface elevation estimate from ASIRAS was evaluated by comparing to the snow surface measured by aggregating laser altimetry observations from the Airborne Laser Scanner (ALS), and the ice surface measured by subtracting ground
observations of snow depth from the snow surface. The perceived penetration of the ice surface estimate was found to increase with the retracker threshold and was correlated with the value of surface properties. The slope of the relationship between penetration and threshold was greater for a deformed ice surface, a rough snow surface, a deeper snow cover, an absence of salinity and a larger snow grain size. As a result, the ideal retracked threshold, one that would achieve $100\%$ penetration, varies depending on properties of the surface being observed. Under conditions such deep snow or a large grain size, the retracked
elevation $s_r$ was found in some cases to not penetrate fully the snowpack. This would cause an overestimation of the sea ice freeboard and as a consequence, the sea ice thickness.

Results suggest that using a single threshold with the TFMRA retracking method will not yield a reliable estimate of the snow-ice interface when observed over an area with diverse surface properties. However, there may be potential to improve the retracking method by incorporating knowledge of the sensed surface physical characteristics. This study shows that remotely
sensed surface properties, such as the ice deformity or snow surface roughness, can be combined with the waveform shape to select an ideal retracker for individual returns with an additional offset to account for the incomplete penetration of Ku-band over appropriate surface characteristics.



# 1 Introduction

The Arctic's annual freeze and melt cycle has a strong influence on ocean and atmospheric currents (Peings and Magnusdottir, 2014; Rosenzweig et al., 2019) and disruption of this cycle can cause shifts to more extreme weather in other parts of the globe (Vihma, 2014). Climatology models have projected a nearly sea-ice free Arctic summer by the mid-21st century, but uncertainty of these models is high due to the incomplete characterization of internal variability of the Arctic climate (Serreze and Meier, 2019) and the omission of multiyear sea ice fluxes (Overland and Wang, 2013). Therefore, the accuracy of projections from models could be improved by the enhancement in accuracy or inclusion of sea ice thickness estimates (Day et al., 2014), as previous models have underestimated the scope of volume loss due to insufficiently dense or accurate measurements of sea ice thickness (Serreze and Stroeve, 2015).

Sea ice thickness can be estimated by applying the principles of hydrostatic equilibrium to measurements of the sea ice height above the ocean (freeboard), the density of sea ice, ocean water and the density and thickness of the snow covering the sea ice (Forsström et al., 2011). The current method of obtaining basin-scale estimates of Arctic Ocean sea ice thickness is to measure the sea ice freeboard using CryoSat-2 and snow depths based on a comprehensive analysis of Arctic climatology (hereafter referred to as W99) which incorporated *in situ* measurements of Arctic snow made between 1954 and 1991 (Warren et al., 1999). Estimates based on the W99 model may no longer be representative of the current Arctic snow cover due to recent changes in the Arctic climate system (Webster et al., 2014). To compensate, bias-corrected snow depth measurements from Operation Ice Bridge (OIB) Arctic campaigns have been used to characterize snow depth over first-year ice (Kwok et al., 2017; Laxon et al., 2013; Kurtz and Farrell, 2011).

CryoSat-2 is a satellite Ku-band (13.5GHz) radar altimeter designed to measure the elevation of the sea ice surface (Gerland et al., 2012), which in turn is used for sea ice thickness estimates (Laxon et al., 2013; Li et al., 2020). In principle, it has been assumed that the scattered radiation used for this measurement is derived from the ice surface and is unaffected by overlying snow accumulation (Laxon et al., 2013; Kwok, 2014). Recent studies have challenged this assumption, suggesting that the dominant response from snow-covered sea ice may originate from within the accumulated snow, likely due to the presence and spatiotemporal variations of snowpack properties, particularly salinity and grain size (King et al., 2018; Nandan et al., 2017b, a). Since sea ice thickness estimates calculated from CryoSat-2 are critical to projections of Arctic sea ice (Tilling et al., 2018), it is important to both quantify and constrain sea ice elevation uncertainties. This can be done by comparing dense measurements of snowpack and sea ice surface properties from *in situ* observations, with radar altimetry measurements captured by an airborne version of the CryoSat-2 SIRAL sensor called ASIRAS. Previous studies have used an airborne platform a laser altimeter (ALS) and ASIRAS to analyze the penetration of the Ku-band signal into the snowpack and have found that the dominant scattering interface trends closer to the snow surface than the expected ice surface (Willatt et al., 2011; King et al., 2018) and that features in the snow freeboard profile were found in the radar signal (Di Bella et al., 2018).

This study presents a detailed analysis into the uncertainties regarding the Ku-band sea ice freeboard estimate, by combining the ALS-ASIRAS dual-instrument airborne platform observations with comprehensive *in situ* observations of multiple surface





properties that are both coincident with the aerial track and cover a distance greater than length-scales of snow depth variability on sea ice. The following objectives are established:

1. Evaluate ASIRAS Ku-band radar altimeter measurements and quantify penetration into the snowpack to constrain the accuracy of the snow-ice interface elevation estimate

60  2. Investigate the effect of surface characteristics on the penetration and accuracy of the snow-ice interface estimate

## 2  Data and Methods

This study compares radar and laser altimetry with on-ice observations collected over land-fast first-year sea ice near Eureka, Nunavut, Canada. Airborne radar data were collected by the European Space Agency during the CryoVEx 2014 Arctic campaign and in situ observations were collected between March 24th, and April 2nd, 2014 by Environment and Climate Change Canada (ECCC) as part of a coordinated field campaign (King et al., 2015).

### 2.1  Airborne Radar and Laser Altimetry

A Twin Otter aircraft carrying the ASIRAS and Airborne Laser Scanner (ALS) flew a path coinciding with the ECCC ground campaign on March 25th 2014, at an altitude of approximately 350 meters and a ground-reference flight speed of $250\text{ms}^{-1}$ (Hvidegaard et al., 2014). The airborne mission produced 66,120 radar and 10,078,030 ALS returns along an approximately 43km long track. The ALS is a 904nm wavelength (near infra-red) Reigl laser scanner with a horizontal resolution of 70cm by 70cm and a vertical accuracy of 10cm (Hvidegaard et al., 2014). The position of the aircraft is measured relative to the WGS84 ellipsoid using a combination of two Global Positioning System (GPS) antennas and an Inertial Navigation System (INS) whose combined positional accuracy is 2cm (Hvidegaard et al., 2014). The ASIRAS altimeter has a vertical range resolution of approximately 10cm, but the primary uncertainty of its measurements comes from the interpretation of the radar waveform return (Ricker et al., 2014).

The ALS swath width was approximately 150m across-track. ASIRAS is a 13.5GHz (Ku-band) radar altimeter with a footprint width of 20m across and 3m along track when flown in low-altitude mode at 300m above ground. The electromagnetic properties of the snowpack surface cause it to scatter the near-infrared wavelength of the laser altimeter, allowing to measure the elevation of the snow surface. The Ku-band pulses from the ASIRAS radar altimeter penetrate the snow, measuring the ice surface elevation. Theoretically, the difference between these surfaces can then be used to obtain the snow thickness over sea ice, which is required to model sea ice thickness (Kurtz et al., 2013). The range of snow surface elevations observed by the ALS was 2.9m, with values between 5.93m to 8.83m above the WGS84 ellipsoid.





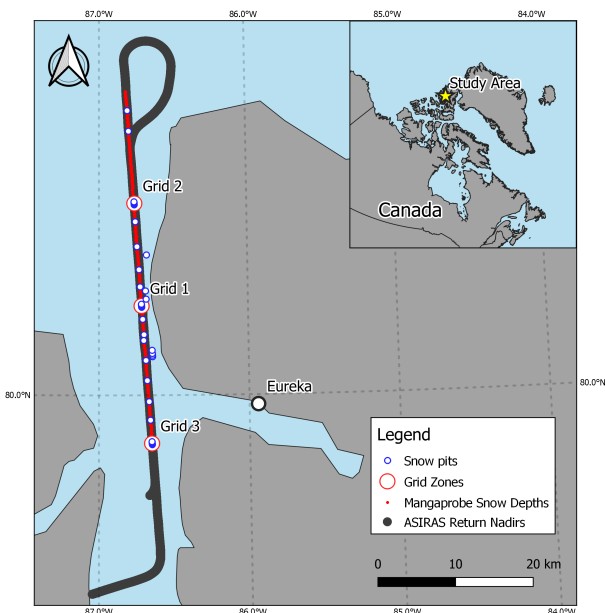

**Figure 1.** Study area showing airborne and field dataset locations. ASIRAS returns collected by the Twin Otter flight are in light grey. Snow depth observations collected by ECCC are in red and snow pits in blue. The research and weather station at Eureka, Nunavut is approximately 13km East of the flight path. Made with data from Natural Earth.

## 2.2 Field measurements

Field measurements were collected by ECCC over a 46km path following the CryoVEx flight line (King et al., 2015). Snow depth measurements were collected using Snow-Hydro Magnaprobes supported with Global Positioning System (GPS) sensors with a horizontal accuracy of 2m (Sturm and Holmgren, 2018). A total of 37,320 collocated measurements were obtained at a spacing of approximately 2m along the flight track, with transects branching out orthogonally at random intervals at lengths of up to 100m. Bulk measurements of snow density were obtained using a ESC30 gravimetric snow sampler at 174 points

along the track, with an average separation of 550m. Approximately 10km of the collection path was coincident with airborne observation track (within the 20m x 3m footprint). The coincident segments of the path ranged in size from 100m to 2860m, with at least 10 segments being over 500m in length. Since a large number of segments are greater than the reported length-scale of snow depth variability on sea ice (King et al., 2015), and span a distance that covers multiple ice floes, the data are likely representative of mid-March conditions on first-year sea ice in the Canadian Arctic Archipelago.

An additional 20,440 measurements were collected from three sites situated approximately 1km, 18km and 31km North of the start of the CryoVEx track. Each site contained a measurement grid of 21 lines covering 500m along the track and 14 lines covering 250m across-track. An additional 54 density measurements were obtained at each site in grids of 3 lines along and 6 lines across-track.





Vertical profiles of the snowpack properties were collected by excavating 37 pits along the course of the track, 14 of which
were within 14 meters of a radar altimeter nadir. A YSI EcoSense EC-300A sensor was calibrated in a control solution of saline
water and used to measured snow salinity with an accuracy of $\pm$ 0.2PSU. Each pit stratigraphy was manually determined by
inspecting the snow pit face with a finger hardness test. Grain size and classifications were determined by selecting 3 grains
deemed to be representative of the layer and measuring their minimum and maximum diameter with a 2mm compactor card
and a field microscope. Nandan et al. (2017b) identified that the snowpack parameters which had the most impact on the Ku-
band microwave penetration were the snow salinity and snow grain size. These properties were measured along vertical profiles
in the snow pits. Ideally, the profile layers were to be aligned with radar altimeter echoes acquired coincidentally with the snow
pit measurements. Due to uncertainties in determining the sensor offset, such an alignment was not feasible. Therefore, the
snowpack properties were summarized per pit as a layer weighted average.

## 2.3 Radar Processing

The ASIRAS return can be described by measuring the pulse peakiness (PP) which is generally calculated as the ratio between
the maximum echo return power and either the sum or the mean of the total echo power. The PP value can be used to indicate
a specular surface scatter (high PP) or a diffuse surface scatter (low PP), and is commonly used to distinguish between open
ocean, leads and sea ice (Peacock and Laxon, 2004; Ricker et al., 2014). While the study only covers landfast first-year sea ice,
the parameterization of the measurements' waveform may be beneficial due to the known effect of surface roughness on the
shape of the waveform (Makynen and Hallikainen, 2009).

The radar waveform was retracked to estimate elevation of the interface within the echo. Estimated elevation was then
compared to the observed elevation of the ice surface, which was obtained by subtracting the field measured snow depth from
the snow surface elevation. A common method for retracking Ku-band sea ice altimeter waveforms to obtain sea ice freeboard
is the threshold first maxima re-tracker algorithm (TFMRA) (Xia and Xie, 2018; Ricker et al., 2014). This method finds a point
on the echo with a value equal to a given percentage (threshold) of the maximum echo value that is the closest such point on
the leading edge of the waveform, which is the left side of the first instance of the echo's maximum value (Helm et al., 2014).
The threshold percentage can vary from 40% to 80% (Ricker et al., 2014), with recent studies suggesting that 70% is the most
effective (Xia and Xie, 2018). To evaluate the effect of the TFMRA threshold selection on the error, thresholds of 20% to 100%
at a 10% interval were applied.

The aircraft carrying the sensor is subject to variations in pitch and roll, causing the target to deviate from the nadir. These
deviations can result in waveform blurring and errors when retracking elevations. Returns associated with roll or pitch deviation
greater than 1.5 degrees were excluded from the analysis (King et al., 2018). Additionally, specific properties of the surface
are considered as a source of uncertainty in the waveform analysis. A parameter called $h_{topo}$ is used as proxy for ice surface
roughness to filter out radar observations in areas of high topographic variability where off-nadir interactions may dominate
the radar response Newman et al. (2014). Estimates of $h_{topo}$ are calculated as the difference between the 95[th] and 5[th] percentile
of snow surface elevations within the footprint.





Due to the positioning and structure of the platform, the ALS and ASIRAS sensors have a relative offset that needs to be calibrated against a common surface elevation. An offset compensation is applied to the ASIRAS radar to align the snow surface return with the ALS measured surface. In the case of CryoVEx, the offset is calibrated over runways (Hvidegaard et al., 2014), using corner reflectors (Willatt et al., 2011), or over open water (King et al., 2018). Since none of these options were available for this study, the sensor offset was calibrated using the distance between the estimated and observed snow-ice interfaces in returns over an area of shallow snow and flat ice. To reduce the effect of snowpack volume scattering on the radar waveform, only returns with an average observed snow depth of less than $10\text{cm}$ within the radar footprint over level ice were considered during calibration.

The elevation of a snow-covered ice surface can be underestimated by radar altimetry due to the the reduced speed of electromagnetic waves in snow (Mallett et al., 2019; King et al., 2020). The retracked elevation is adjusted by applying a correction factor such that the speed of light through the snow is given by

$$c_{snow} = \frac{c_{vacuum}}{\sqrt{1+2p_s}} \tag{1}$$

where $p_s$ is the density of the snow in $\text{gcm}^{-3}$ (Kurtz et al., 2013).

The retracked elevation $s_r$ for each ASIRAS return is compared to the elevation of the snow and ice surface within the footprint of that return. The snow surface $s_a s$ is obtained from the average elevation of ALS points within the footprint, while the ice surface $s_s i$ is calculated by subtracting the average of ground-based snow depth observations within the footprint from the snow surface. To evaluate the ability of the Ku-band radar altimeter to measure the elevation of the snow-ice interface, penetration ($P$) and the accuracy of the interface estimate ($E$) are calculated. Estimates of $P$ are retracked elevation relative to the snow surface elevation, with positive being below and negative being above. Quantities of $E$ are retracked elevation relative to the ice surface elevation, with positive being above and negative being below. The penetration can also be expressed as a proportion of the observed snow depth, which yields the percentage of the snowpack penetrated $P_r$ or proportion of the snowpack as error $E_r$. To measure the magnitude of error over a group of returns, the absolute values of $E$ and $E_r$ are expressed as $E_a$ and $E_{ra}$ respectively.

The sensor used in the ASIRAS radar altimeter is based on the SIRAL sensor design used in Cryosat 2, so its footprint can be estimated using the same principles (Mavrocordatos et al., 2004). For the tracks used in this study, the ASIRAS altimeter was flown in single-antenna SAR, low-altitude mode. Due to the nadir-facing orientation of the sensor, the footprint is pulse-doppler-limited (PDL), and based on its flight parameters has an approximate size of 2 meters in the along-track and 20 meters in the across track direction (Scagliola, 2013). Measurement of the surface such as snow depth, surface elevation and ice deformity classification are aggregated to the footprint for each return to estimate the surface characteristics observed by the radar altimeter. To evaluate the best method for aggregating measurements, circular areas with radii ranging from 8 to 40 meters at 2-meter intervals are also used as aggregation footprints.



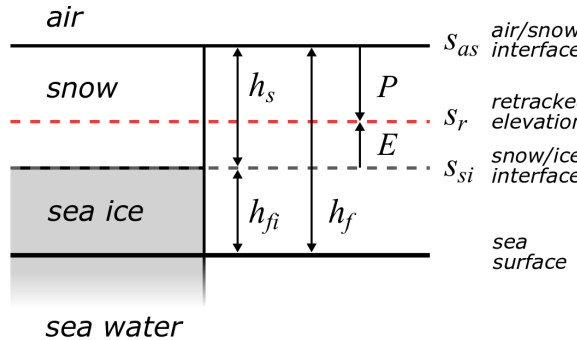

**Figure 2.** Schematic of a simplified snow on sea ice model showing the physical representation of variables used to calculate the radar altimeter penetration $P$ and ice surface estimate error $E$. Unlabeled variables are the height of the snowpack $h_s$, snow freeboard $h_f$, and the ice freeboard $h_{fi}$.

## 3   Results and Analysis

To evaluate the accuracy of the retracker-estimated ice surface elevation, the footprint used to aggregate measured snow surface
properties must represent the area of the surface observed by ASIRAS. Ideally, this is the pulse-doppler limited (PDL) footprint, but rough surfaces can increase the distance from nadir at which features appear in the radar signal (Newman et al., 2014). In the past larger circular footprints have been used to represent the ASIRAS footprint and compensate for off nadir influence (King et al., 2018).

In Figure 3, the mean and standard deviation of the error $E_{ra}$ of all ASIRAS returns is plotted for multiple footprint radii.
It shows the magnitude and variability of the error $E_{ra}$ change with the radius of a circular footprint, reaching a minimum at approximately 14 meters. The error of $E_{ra}$ for the circular footprint (blue line) is lower for almost all radii than for the PDL (black line at $44.9\%$ and $34.7\%$ respectively). Since the number of surface observations aggregated by a footprint increases with radius, it is expected that the error decreases with size. The increasing error beyond 14 meters suggests that the footprint covers almost all of the surface features that contribute to the signal, and that any features beyond are not correlated to the return.
This effect may be due to the roughness of the ice surface, where ice features that are elevated above the surface at the nadir can appear in the radar signal at distances proportional to their height, but can also be affected by the spatial autocorrelation of surface properties. A 14-meter radius footprint is used henceforth in this analysis to improve the reliability of the snow and ice surface reference elevations, and as a best-case scenario when evaluating the retracker accuracy, to reduce the contribution the footprint size to the retracker error.
In Figures 4 and 5, the penetration as a proportion of the snow depth $P_r$ is used to demonstrate the retracked elevation in relation to the snow and ice surfaces, but the same effect is seen in the penetration $P$ and error $E$ measurements. The position of the retracked elevation relative to the snow and ice surfaces follows a normal distribution whose mean and variability are affected by the retracker threshold (Figure 4). Selecting a larger retracker threshold increases the penetration of the retracked





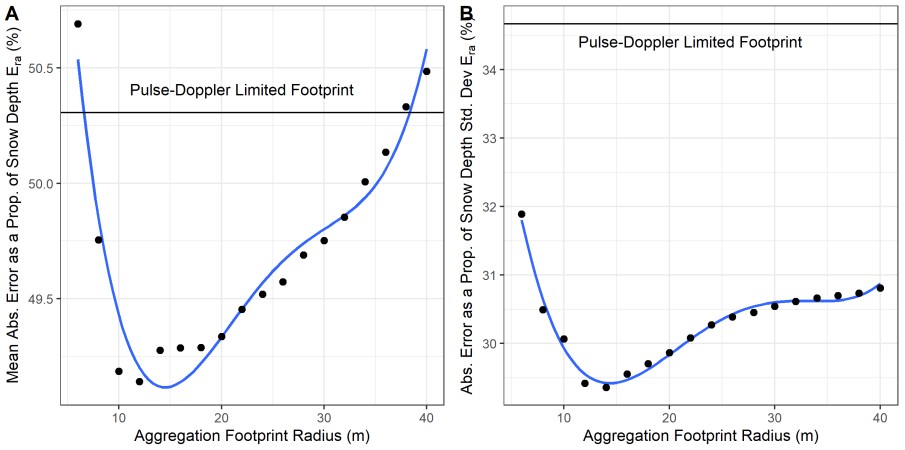

**Figure 3.** Mean (A) and standard deviation (B) of the absolute error relative to snow depth $E_{ra}$ across different radii for the aggregation footprint, fitted with $4^{th}$ degree polynomial curve. The value for the pulse-dopper limited footprint is shown as a black horizontal line.

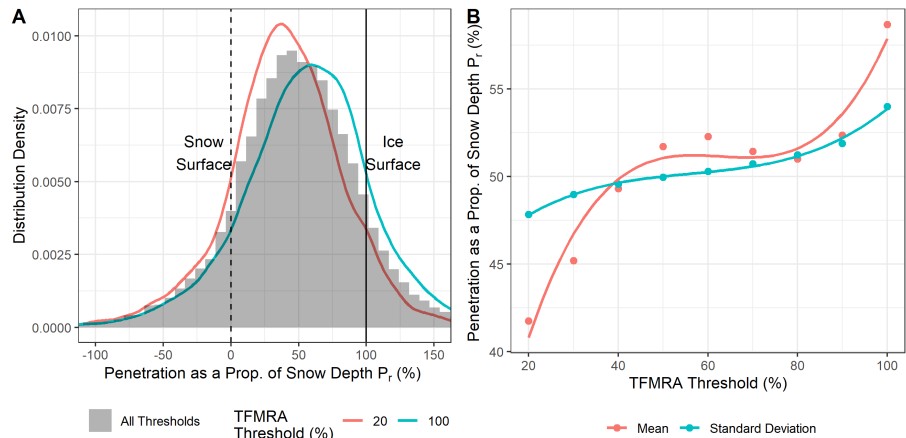

**Figure 4.** (A) Distribution of values of penetration through the snowpack $P_r$ relative to the observed snow (marked $0\%$) and ice surface (marked $100\%$). Grey histogram represents penetration across all retracker thresholds while the PDF curves represent the distribution for two select retracker thresholds. (B) Mean and standard deviation of the penetration $P_r$ over retracker threshold fitted with a $3^{rd}$ degree polynomial.

elevation into the snowpack, with the lowest retracker threshold ($20\%$) having a $P_r$ of $42\%$ and the highest ($100\%$) having a
value of $59\%$. A higher threshold not only moves the retracked elevation distribution closer to the ice surface, but also increases
its variability, resulting in more retracked elevations to appear below the ice surface (from $8\%$ to $16\%$) and fewer to appear
within the snowpack (from $77\%$ to $73\%$).

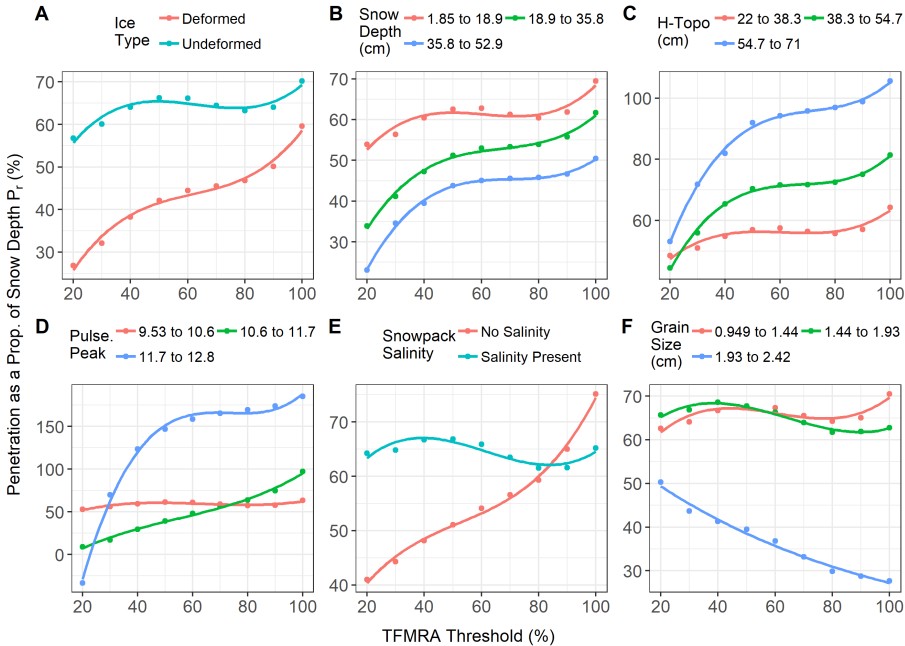

**Figure 5.** The mean penetration through snowpack $P_r$ over TFMRA threshold for categories of ice type (A), snow depth (B), h-topo (C), snowpack salinity (D), grain size (E) and pulse peakiness (F). Plots A-D summarize 22,030 ASIRAS returns that had surface observations within their footprint, while plots E and F only summarize 247 returns that had a snow pit within their footprint.

Mean penetration $P_r$ increases with retracker threshold in a relationship that approximates the logit function (Figure 4B), which is an expected result given that leading part of the radar altimeter waveform can be fitted using a Gaussian curve (Zygmuntowska et al., 2013). An ideal threshold would be one where the average retracked elevation has $100\%$ penetration through the snowpack, but Figure 4B shows that the average penetration does not exceed $56\%$, even at the waveform peak ($100\%$ threshold).

To investigate how surface properties can affect the penetration ($P_r$) and its relationship with the retracker threshold, penetration is shown for categories of ice deformed type, snow depth, $h_{topo}$, pulse peakiness, snowpack salinity and snow grain size in figure 4. The value of these surface properties appears to affect both the average penetration and the slope of the penetration-threshold relationship. The $P_r$ for all categories of $h_{topo}$ starts at approximately $50\%$ ($20\%$ threshold), but the high $h_{topo}$ returns gain a $50\%$ penetration (to the $100\%$ threshold), while the low $h_{topo}$ returns only gain $20\%$ (Figure 5c). In contrast, the shallowest snow depth category only gains $15\%$ compared to the other categories' $25\%$, but the average penetration increases by $10\%$ to $15\%$ across progressively shallower categories. Among the returns that had a snow pit within their footprint, those with no snowpack salinity showed a strong increase of penetration with threshold, while those with salinity had an almost constant penetration at 60 to $70\%$ (Figure 5). A smaller grain size from $0.949$mm to $1.93$mm showed a consistent penetration, but larger grain sizes had the only observed negative penetration-threshold slope.





## 4   Discussion

The primary issue highlighted by the analysis is that a TFMRA retracker alone could not be used to obtain an accurate es-
timate of the ice surface elevation for FYI near Eureka. The method used to calibrate the ALS-ASIRAS sensor offset uses
snow-covered ice, which should underestimate the offset, resulting in a lower retracked elevation and an overestimate of the
penetration. Given that systematic errors of the analysis favour a higher penetration, the peak penetration $P_r$ of $57\%$ through
the snowpack suggests that the Ku-band radar signal is not consistently returning from the ice surface, but from within the
snowpack. Since the sea ice thickness is estimated from the sea ice freeboard and snow depth using a hydrostatic equilibrium
model (Tilling et al., 2016; Li et al., 2020), the error between the retracked elevation and the actual ice surface would be mag-
nified in the ice thickness error by a proportion relative to the ratio of the sea water density to the snow density. To accurately
measure the ice surface elevation under the conditions of this study, the TFMRA retracker would need to be augmented with
an offset that would correct for the distance between the retracked elevation and the ice surface.

A method that extends the penetration of the retracked elevation would also need to account for the difference the in
penetration-threshold between observations with different surface properties. The ideal retracker threshold for a surface with a
high surface roughness or deformed ice surface might be lower than one for a low-$h_{topo}$ surface or an undeformed surface. In
this case there is potential to improve the radar altimetry by incorporating surface properties measured by other remote sensing
platforms (Landy et al., 2020). The ice deformity type was extracted from the C-band RADARSAT-2 while the $h_{topo}$ calculated
from ALS observations. CryoSat-2 observations may then be assisted by measurements from the RADARSAT Constellation
Mission (RCM), ICESat-2 or by other platforms which can give insight into characteristics of the Arctic snowpack and ice
surface.

The shape of the radar waveform may also have potential in augmenting the TFMRA method, as it is correlated with
differences in the penetration-threshold relationship. At the leading edge of the waveform peak, the strength of the return
signal increases with distance from the sensor. A point along the leading edge at a higher threshold of the peak will be closer
to the peak, and will have penetrated further (deeper) into the snowpack, resulting in a positive relationship between retracker
threshold and penetration. A higher threshold-penetration slope, as seen in Figure 5, is likely caused by a wider and flatter
return signal with a lower pulse peakiness. In Figure 5d, the pulse peakiness has an effect on the relationship slope similar
to that of $h_{topo}$ and an effect on the average penetration similar to that of show depth. It is possible, therefore, that the pulse
peakiness can act as a proxy for a combination of the surface roughness and snow depth, and can be used to approximate the
effect of those characteristics on the penetration.

The results of this study are consistent with the findings of Nandan et al. (2017a), who determined that snow depth and
salinity can impact the penetration of Ku-band radar through snow, potentially overestimating the ice surface elevation and
reducing the accuracy. They suggested that a correction factor based on distributed measurements of snow depth and salinity can
be applied to the freeboard to improve accuracy. Although Guerreiro et al. (2017) used pulse peakiness to correct the freeboard
estimates of the Ku-band radar altimeter RA-2 aboard Envisat, no studies have investigated the potential for correcting CryoSat-
2 freeboard estimates using information from the radar altimeter waveform. Since a simple measure of the waveform shape such





as pulse peakiness is shown to have similar effects on the penetration-threshold relationship as physical surface characteristics, there may be potential to use more complex interpretations of the waveform to more appropriately select a TFMRA retracker threshold and improve the freeboard estimate.

## 5 Conclusions


The objectives of this study were to evaluate the ASIRAS Ku-band radar ice surface estimate accuracy, penetration into the snowpack, and investigate how these are affected by surface properties and explore methods which could improve the accuracy of the elevation estimate. Over the study area, the retracked elevation was found to penetrate $50\%$ through the snowpack on average, with a standard deviation of $51\%$. The average penetration and its variability increased with the retracker threshold,

and were correlated with changes in surface properties such as ice deformity, snow depth and $h_{topo}$, and snow pit measured properties, particularly snowpack salinity and snow grain size. The retracked elevation could not consistently estimate the ice surface elevation, likely due to the incomplete penetration of the radar through the snowpack. In the context of the study area, over landfast first-year sea ice, the TFMRA alone may not be sufficient to estimate the ice surface elevation from the Ku-band radat altimeter. There is potential to augment the TFMRA retracker by using the waveform shape (characterized by the pulse

peakiness) and surface characteristic estimates from other remote sensing platforms to calculate an offset which supplements the lack of penetration or calculate an ideal retracker threshold in areas where the Ku-band radar is able to penetrate the snowpack. Improvements in the accuracy of the sea ice surface elevation are crucial, since a small overestimate of the sea ice freeboard can result in larger overestimates of the sea ice thickness based on the hydrostatic equilibrium model.

## 6 Code and data availability

Snow and ice data from the ECCC field campaign is available for download through the Government of Canada Open Data Portal (https://doi.org/10.5281/zenodo.823679). Airborne radar data evaluated as part of this study and collected during the CryoVEx ASIRAS Campaign (https://doi.org/10.5270/esa-aa4xtkn) is available for download through the ESA EO Campaign Portal (https://earth.esa.int/web/guest/pi-community/apply-for-data/campaigns). Code and instructions to reproduce all figures and analysis is available in an open access repository (https://github.com/pjdon/cveureka)

*Author contributions.* P.D wrote the manuscript with input from J.K. and R.K. P.D. developed and preformed the radar analysis. J.K. coordinated and preformed the field measurement campaign. All authors contributed to the experiment design

*Competing interests.* The authors declare no competing interests.



*Acknowledgements.* The authors would like to thank the Polar Continental Shelf Program and Eureka Weather Station for their logistical support of this work. The Eureka field campaign was funded by Environment and Climate Change Canada and the Canadian Space Agency
through the Government Related initiatives Program (GRIP). We thank Stephen Howell, Arvids Silis, Peter Toose, and Nick Rutter for their efforts during the field campaign. Thank you to the ESA and DTU flight and science teams for their efforts to coordinate data collection at Eureka. The authors also acknowledge the support of the Natural Sciences and Engineering Research Council of Canada (NSERC) [funding reference number xxxxxx]





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
