# Peer review of "Evaluating Airborne Ku-Band Radar Altimetry over Landfast First-Year Sea Ice"

_The Cryosphere, 2020_

## Referee Comment (RC1) · Anonymous Referee #1 · 12 Jan 2021

Review Evaluating airborne Ku-band radar altimetry over landfast first-year ice

Major comments:

The paper evaluates Ku-band radar penetration into snowpack acquired from ASIRAS system. Field datasets were acquired from ECCC coordinated field campaign near Eureka in 2014. The study is timely given the growing need for more reliable sea ice thickness estimates from ongoing and upcoming radar missions. The methodology of the study is very thorough. However, the paper is not well written for a scientific research article to be published.

The biggest concern I have is that this paper lacks sufficient literature review for a research article. Details on previous key studies on radar altimeter and its uncertain-

[Figure]

ties due to snow variability need to be elaborated. Gaps and challenges in previous altimeter retracking methods need to be addressed to justify the research objectives.

Fig. 5 demonstrated mean penetration variability for different snow surface characteristics and explained in line 193-203. I do not think the authors can not comment straight forward on penetration variability just based on surface characteristics where other variables are already affecting the penetration variability. A sensitivity study is required to demonstrate the contribution of individual surface characteristics to penetration depth. Also, a discussion on the partial contribution of other snow properties is missing in the discussion section.

This discussion needs to be revised with a focus on analysis results and geophysical explanation of observed variability of penetration. A thorough discussion is required. The discussions should be supported by previous studies, where only Landy 2020 was mention without proper discussion. I suggest 'Major revision' for this paper.

Minor comments:

Line 15 ". . . in some case to not penetrate. . .'. Please rewrite. Fig 1 'the Twin Otter flights are in light grey' – is confused with grey land area. Try different color codes for flight lines, or rewrite the captions to clarify the flight lines and land area. What is Natural Earth? Provide a reference or weblink for natural Earth. Please use a larger font for coordinates. Line 138 How the level ice area was defined? Were there any quantifications of the surface roughness? Line 147-148 The acronyms do not correspond to Fig 2. Check throughout the manuscript. Line 164-168 This section belongs to the method section. Line 214 ". . . . the in.." should be 'in the'? Line 214-221 This section is more like an outlook, which should be in the conclusion. Line 249 Should be 'radar'

---

## Referee Comment (RC2) · Anonymous Referee #2 · 1 Feb 2021

Dear Authors

The following comments reflect my observations on the TC manuscript "Evaluating Airborne Ku-Band Radar Altimetry over Landfast First-Year Sea Ice" by Donchenko, King and Kelly. The study focused on quantifying the impact of snow properties and surface characteristics, and its impact on the accuracy and penetration of the TFMRA retracking algorithm, operationally used for sea ice thickness estimations from radar altimetry.

This study is relevant to the scientific community with respect to improving our understanding of how snow critically impacts the accuracy of satellite radar altimeter-derived sea ice thickness estimates from presently operational and forthcoming radar altimeters such as CryoSat-2, ALtiKa and CRISTAL missions.

[Figure]

Although the study focuses on the uncertainties, after thoroughly reading the paper, I was left with even more uncertainties and research gaps (mainly due to unclear methodology and lack of rigorous analysis and discussion). The manuscript, as it stands, reads 'incomplete', and requires significant major addition and revision, before it can be deemed worthy of publication in TC.

I am willing to review the revised manuscript once all my overall general comments are addressed. I would like to keep the review short to addressing major concerns in the manuscript, before delving into the editorial comments (probably in the next round of review). Please see my general comments below, which covers my major concerns.

a) My main issue is Figure 5 (which is the corner stone of this study, correct?), and how you arrived at Figure 5 (no methods mentioned), what data was used, and no proper discussion (or even analysis) of the different snow/surface parameters you used to correlate with the TFMRA thresholds. a1) Although, the authors do vaguely mention about the different snow pits dug during their campaign, there is no explanation of these snow pits where you need to demonstrate how the different snow covers looked like in terms of vertical profiles of properties such as density, temperature, salinity, grain size etc. How are the snow pits spatially different? Were snow pits cold or warm? (because snow temperature also affects the radar penetration, especially at high frequencies correct?). Were the snow pits vertically homogeneous? (any layering or ice lenses present?). You mention snow salinity as a critical factor affecting the retrievals. Were the snow salinity profiles typical of FYI?

In summary, since this study depends heavily on the snow properties, the authors needs to spend time to decipher your snow pits.

a2) On that note as a follow up to a1), if your snow pits are diverse, then I am curious to learn how the radar penetration proportion through snow is affected by the variability in the snow properties. Yes, you do show the range in Figure 5. But I cannot believe the numbers. For example, if you look at the grain size estimates, from a range of 1.93

cm to 2.42 cm (which is insanely high), I didnt see a standard deviation. For a high frequency such as Ku-band, I cannot believe (sorry) that the penetration is the same for these two extreme snow grain sizes. The same is applicable for snow salinity. You labelled in the plot as 'no salinity' and 'salinity present'. I think this information is very vague. 'salinity present' can be 0.00001 psu/ppt correct? Linking back to a1), the authors need to be clearly mention the range of these properties and clearly demonstrate the sensitivity of these properties.

a3) I am not clear about how (more important) and why the authors used a layer weighted average (Line 108) method to summarize the pits. That instantly shows the flaw in your analysis, which needs to be rectified (linking a1 and a2).

b) My second problem is the methods section. There is no description or flowchart of how the observations and modeling was employed. This causes the reader to go clueless about how the results and analysis were conducted.

b1) The authors has already done a good job introducing the radar measurements. However, they need to showcase how the snow properties were used to interpret observations, and hence produce Figure 5 and its analysis.

b2) It would b great if the authors can show the readers how the environment (sea ice) and distinct snow pits looked like (they mention about snow pit faces?).

c) My next concern for now is the discussion section. From my initial reading of this section, it seems the authors blindly recommends improvements to be made for TFMRA, without a proper discussion of the results. I see this section more as a 'Future recommendations' section than a discussion of the results. With a significant room for increasing the word limit (in regards to page length), the authors should refocus on the analysis and presentation of results (linking a)) and then discuss the results and also MENTION the study limitations also.

d) My last concern is definitely the introduction section. Its short (that's still fine with

me), but lacks guiding the reader through the key concepts (including the role of snow in sea ice thickness retrievals) and particularly focusing on the ambiguities and uncertainties. Yes, the authors do talk about it, but only vaguely. I am also curious to see a literature review (brief) of how different snow properties and ice types (not just FYI) affect the ice thickness retrievals. That will anyways help the authors to have a strong platform from where, they can introduce the rationale, research gaps and objectives.

---

## Editor Comment (EC1) · Lars Kaleschke (Editor) · 2 Feb 2021

Dear authors,

two referees have provided substantial comments on your manuscript. On the one hand it was acknowledged that the study is timely and relevant to improve our understanding of how snow critically impacts the accuracy of satellite radar altimeter-derived sea ice thickness estimates. On the other hand both referees agree that the paper is not well written for a scientific research article and that it reads 'incomplete'. Both referees ask for major revisions and I agree that this is necessary until the manuscript could be properly evaluated. Because of the special shortness and the shortcomings I was tempted to reject the manuscript at the current stage. However, given that both

referees are willingly to review the manuscript a second time I encourage the authors to consider the constructive and detailed comments for a next round.

Best regards

Lars Kaleschke